# Surgery of Colorectal Liver Metastases Involving the Inferior Vena Cava: A Systematic Review

**DOI:** 10.3390/cancers15112965

**Published:** 2023-05-29

**Authors:** Mario Serradilla-Martín, José Ramón Oliver-Guillén, Pablo Ruíz-Quijano, Ana Palomares-Cano, Roberto de la Plaza-Llamas, José Manuel Ramia

**Affiliations:** 1Department of Surgery, Instituto de Investigación Sanitaria Aragón, Hospital Universitario Miguel Servet, 50009 Zaragoza, Spain; mserradilla@salud.aragon.es; 2Department of Surgery, Hospital Santa Bárbara, 42005 Soria, Spain; 3Department of Surgery, Hospital HM Sanchinarro, 28050 Madrid, Spain; pruizquijano@hmhospitales.com; 4Department of Surgery, Hospital Universitario Miguel Servet, 50009 Zaragoza, Spain; apalomaresc@salud.aragon.es; 5Department of Surgery, Hospital Universitario de Guadalajara, 19002 Guadalajara, Spain; rdplazal@sescam.jccm.es; 6Department of Surgery, Hospital General Universitario Dr. Balmis, 03010 Alicante, Spain; ramia_jos@gva.es

**Keywords:** colorectal neoplasms, neoplasm metastasis, liver, venae cava, hepatectomy, treatment outcome

## Abstract

**Simple Summary:**

Most of the existing data on the resection of liver metastases with inferior vena cava infiltration come from case reports or small case series. This systematic review on the subject found postoperative morbidity and mortality rates and overall survival to be acceptable, offering a surgical alternative to these patients previously considered unresectable.

**Abstract:**

Combined hepatic and inferior vena cava (IVC) resection is the only potentially curative treatment for patients with colorectal liver metastases (CRLM) involving the IVC. Most of the existing data come from case reports or small case series. In this paper, a systematic review based on the PICO strategy was performed in accordance with the PRISMA statement. Papers from January 1980 to December 2022 were searched in Embase, PubMed, and the Cochrane Library databases. Articles considered for inclusion had to present data on simultaneous liver and IVC resection for CRLM and report surgical and/or oncological outcomes. From a total of 1175 articles retrieved, 29, including a total of 188 patients, met the inclusion criteria. The mean age was 58.3 ± 10.8 years. The most frequent techniques used were right hepatectomy ± caudate lobe for hepatic resections (37.8%), lateral clamping (44.8%) for vascular control, and primary closure (56.8%) for IVC repair. The thirty-day mortality reached 4.6%. Tumour relapse was reported in 65.8% of the cases. The median overall survival (OS) was 34 months (with a confidence interval of 30–40 months), and the 1-year, 3-year, and 5-year OS were 71.4%, 19.8%, and 7.1%, respectively. In the absence of prospective randomized studies, which are difficult to perform, IVC resection seems to be safe and feasible.

## 1. Introduction

Hepatic resection is the only potentially curative treatment for metastatic and primary tumours of the liver, with 5-year survival rates of up to 50% for colorectal liver metastases (CRLM) [1,2,3,4]. Today, chemotherapy is not a curative option as the sole treatment, with a median survival of less than 12 months if these patients are left untreated [5].

Classically, patients with CRLM involving the inferior vena cava (IVC) were not considered surgical candidates, with poor overall survival rates. The suspected involvement of the IVC was considered a contraindication for surgery due to the high risk of massive intra-operative bleeding or air embolisms [6].

The continuous development of surgical innovations, such as total hepatic vascular exclusion (THVE), venovenous bypass, and ex vivo hepatic resection, combined with new chemotherapy regimens, has allowed the performing of combined hepatic and IVC resections. For this reason, the patient candidates for surgery have increased, with the publishing of more and more favourable data regarding clinical outcomes [7,8,9,10,11]. 

However, the number of articles published on the subject, and the number of patients described, are low. For this reason, drawing conclusions about the value of this procedure is difficult. Most of the existing data come from case reports or small case series emphasizing surgical innovations, and the long-term results and oncological outcomes are still confusing.

This paper is a systematic review of the literature for the purpose of collecting data from patients who underwent surgery for CRLM involving IVC and evaluating the safety, clinical, and oncological long-term outcomes.

## 2. Materials and Methods

A systematic review was performed based on the PICO strategy [12], following the recommendations of the Preferred Reporting Items for Systematic Reviews and Meta-Analyses (PRISMA) statement [13]. According to the Oxford Centre for Evidence-Based Medicine’s levels of evidence [14], the level of evidence of each study was described. This review is registered at www.researchregistry.com (accessed on 11 April 2023), with the unique identification number: reviewregistry1579. 

### 2.1. Literature Review

A systematic literature search was conducted using Embase, PubMed, and the Cochrane Library to identify relevant articles published from January 1980 to December 2022. The search terms used in Embase were *(‘venae cavae’/exp OR ‘venae cavae’ OR (venae AND cavae) OR ‘cavae, venae’ OR (cavae, AND venae) OR ‘vena cava’/exp OR ‘vena cava’ OR ((‘vena’/exp OR vena) AND cava) OR ivc) AND ((‘neoplasm metastasis’/exp OR ‘neoplasm metastasis’ OR ((‘neoplasm’/exp OR neoplasm) AND (‘metastasis’/exp OR metastasis)) OR ‘neoplasm metastases’ OR ((‘neoplasm’/exp OR neoplasm) AND (‘metastases’/exp OR metastases)) OR ‘metastases, neoplasm’ OR ((‘metastases,’/exp OR metastases,) AND (‘neoplasm’/exp OR neoplasm)) OR ‘metastasis, neoplasm’ OR ((‘metastasis,’/exp OR metastasis,) AND (‘neoplasm’/exp OR neoplasm)) OR metastase OR ‘metastases’/exp OR metastases OR ‘metastasis’/exp OR metastasis) AND (‘colorectal neoplasms’/exp OR ‘colorectal neoplasms’ OR (colorectal AND (‘neoplasms’/exp OR neoplasms)) OR ‘colorectal neoplasm’/exp OR ‘colorectal neoplasm’ OR (colorectal AND (‘neoplasm’/exp OR neoplasm)) OR ‘neoplasm, colorectal’ OR ((‘neoplasm,’/exp OR neoplasm,) AND colorectal) OR ‘neoplasms, colorectal’ OR ((‘neoplasms,’/exp OR neoplasms,) AND colorectal) OR ‘colorectal tumors’ OR (colorectal AND (‘tumors’/exp OR tumors)) OR ‘colorectal tumor’/exp OR ‘colorectal tumor’ OR (colorectal AND (‘tumor’/exp OR tumor)) OR ‘tumor, colorectal’/exp OR ‘tumor, colorectal’ OR ((‘tumor,’/exp OR tumor,) AND colorectal) OR ‘tumors, colorectal’ OR ((‘tumors,’/exp OR tumors,) AND colorectal) OR ‘colorectal cancer’/exp OR ‘colorectal cancer’ OR (colorectal AND (‘cancer’/exp OR cancer)) OR ‘cancer, colorectal’ OR ((‘cancer,’/exp OR cancer,) AND colorectal) OR ‘cancers, colorectal’ OR ((‘cancers,’/exp OR cancers,) AND colorectal) OR ‘colorectal cancers’ OR (colorectal AND (‘cancers’/exp OR cancers)) OR ‘colorectal carcinoma’/exp OR ‘colorectal carcinoma’ OR (colorectal AND (‘carcinoma’/exp OR carcinoma)) OR ‘carcinoma, colorectal’/exp OR ‘carcinoma, colorectal’ OR ((‘carcinoma,’/exp OR carcinoma,) AND colorectal) OR ‘carcinomas, colorectal’ OR (carcinomas, AND colorectal) OR ‘colorectal carcinomas’ OR (colorectal AND carcinomas)) OR ‘colorectal liver metastases’/exp OR ‘colorectal liver metastases’ OR (colorectal AND (‘liver’/exp OR liver) AND (‘metastases’/exp OR metastases))) AND (‘liver’/exp OR liver OR livers).* The search terms used in PubMed were: *((Venae Cavae) OR (Cavae, Venae) OR (Vena Cava) OR (IVC)) AND (((Neoplasm Metastasis) OR (Neoplasm Metastases) OR (Metastases, Neoplasm) OR (Metastasis, Neoplasm) OR (Metastase) OR (Metastases) OR (Metastasis)) AND ((Colorectal Neoplasms) OR (Colorectal Neoplasm) OR (Neoplasm, Colorectal) OR (Neoplasms, Colorectal) OR (Colorectal Tumors) OR (Colorectal Tumor) OR (Tumor, Colorectal) OR (Tumors, Colorectal) OR (Colorectal Cancer) OR (Cancer, Colorectal) OR (Cancers, Colorectal) OR (Colorectal Cancers) OR (Colorectal Carcinoma) OR (Carcinoma, Colorectal) OR (Carcinomas, Colorectal) OR (Colorectal Carcinomas)) OR ((colorectal liver metastases))) AND ((Liver) OR (Livers)).* The search term used in the Cochrane Library was *inferior vena cava.* Searched bibliographies of the retrieved studies were checked to collect additional studies.

Four authors reviewed the studies retrieved that met the inclusion criteria. Possible discrepancies between all the authors were resolved by discussion and consensus among all of them. A cluster search identified some additional papers.

### 2.2. Inclusion Criteria

Papers considered for inclusion had to (a) have patients with simultaneous liver and IVC resection for CRLM; (b) publish original data; (c) report patients’ surgical and/or oncological outcomes.

### 2.3. Exclusion Criteria

(a) Non-English language articles; (b) papers without original data; (c) IVC resection for any type of tumour other than CRLM; (d) duplicate series.

### 2.4. Variables Studied

The data compiled included the following variables for the study: (a) journal and year of publication; (b) demographic variables and characteristics of the patients: country of origin, number of patients included, mean age in years, sex, the mean number of metastatic lesions, and size of the closest lesion to the IVC; (c) surgical variables: type of hepatectomy, type of vascular control, size of resected IVC, type of IVC reconstruction (including size and origin of the autologous patch, if used), combined resection of other visceral structures, and operative time; (d) morbidity and mortality results, according to the Clavien–Dindo classification [15] when possible, and hospital stay; (e) oncological outcomes, such as overall survival (OS) and disease-free survival (DFS), pathological infiltration of IVC, state of margins, disease recurrence and metastasis location, and re-do surgeries, if performed. Individual patient survival was described in some case series; this information was extracted and studied using a Kaplan–Meier analysis.

## 3. Results

The literature search yielded 1175 articles (Figure 1), of which 42 were assessed for eligibility. Thirteen of these papers were excluded: eight were written in a language other than English, and five did not present the original information. A total of 188 patients were described in the 29 articles that met the inclusion criteria [16,17,18,19,20,21,22,23,24,25,26,27,28,29,30,31,32,33,34,35,36,37,38,39,40,41,42,43,44,45]. All the studies were observational and were classified as level-4 evidence, according to the Oxford Centre for Evidence-Based Medicine’s levels of evidence [14]. The characteristics of the study population are described in Table 1 and Table 2. Some data fields could not be fulfilled.

### 3.1. Patients and Surgical Characteristics

A total of 188 patients with combined liver and IVC resection for CRLM were initially included (Table 1). The mean age was 58.3 ± 10.8 (*n* = 140), and 52.3% of the patients were men (79/151). The mean number of resected metastases was 2 ± 3.8 (*n* = 68). The mean metastasis size was 61 ± 53 mm (*n* = 80). The most frequent hepatic resections were right hepatectomy ± caudate lobe (CL) resection (37.8%; 51/135), right trisectionectomy ± CL resection (14.8%), extended right hepatectomy ± CL resection (14.1%), wedge resection (10.4%), segmentectomy (6.7%), and CL resection (4.4%). Combined organ resections mainly included the diaphragm (five cases), right adrenal gland (three), diaphragm and right adrenal gland (two), and right kidney (one). The most commonly used vascular control techniques were lateral clamping (44.8%; 52/116), THVE (28.4%), and IVC clamping (6%). Some authors described the vascular control length. IVC clamping was used for 12–15 min in two cases [21], 17 min in one [17], and 55 min in one [22]. The vascular bypass times were [21] 210–280 min for portosystemic venovenous bypass (two cases) [21], 190–210 min for portosystemic venovenous bypass with hepatic vascular exclusion (HVE) (two cases), and 45–55 min for systemic venovenous bypass with HVE (two cases). THVE was used for 13.6 min (six cases) [43] and extracorporeal circulation for 132 min and up to 163 min (two cases) [37].

The most frequently used technique for IVC repair was primary closure (PC) (56.8%; 92/162), followed by a polytetrafluoroethylene (PTFE) tube graft (23.5%), a PTFE patch (5.6%), a Dacron^®^ patch–tube (3.1%), and an autologous patch graft (7.4%). The autologous patch grafts were used in the following sites: the pericardium (four cases), middle hepatic vein (MHV) (three), saphenous vein (two), external iliac vein (one), ovarian vein (one), and IVC (one).

The laparoscopic approach was performed in one case [42]. The mean operative time was 431.2 ± 167.6 min (*n* = 26).

### 3.2. Morbidity and Mortality Outcomes

Half of the patients developed complications of some kind (52%; 25/48). A blood transfusion was not considered a complication, and it was required in 58.8% (20/34). Clavien–Dindo classification [15] could only be applied to thirty-four patients, ranging from grade 0 in fourteen patients, grade 1 in nine, grade 3a in four, to grade 5 in seven. The grade 3a complications included three cases of pleural effusion that required surgical drainage [21] and one case that required continuous venovenous hemofiltration and endovascular treatment because of a renal artery stenosis [37]. The grade 5 complication (in-hospital mortality) was specified in seven patients [21,27,35,36,39]. The causes were liver failure in three cases [27,35,36], multi-organ failure in another two cases [21,35], hemopneumothorax, renal, and respiratory failure in one case [21], and pulmonary embolism in the remaining case [39]. The mean hospital stay was 15.1 ± 12 days (*n* = 31). The thirty-day mortality was 4.1% (7/169). The overall mortality rate (including in-hospital mortality) rose by 48.2% (66/137). A total of 46 patients (69.7%) died of disease progression. Two patients died with evidence of disease [36,37]. No information could be obtained on the remaining eleven patients. 

### 3.3. Pathological, Oncological, and Survival Outcomes

Histological infiltration of the IVC was demonstrated in 52.6% of the cases (40/76). The margins were infiltrated by tumours in 10.5% (9/86).

Tumour relapse was recorded in 65.8% of the cases (73/111). The most commonly affected organ was the liver (35.4%; 17/48), followed by the lung (16.7%) and lymph nodes (10.4%). The mean time of relapse was 8.5 ± 7.7 months (*n* = 31). Combined organ involvement was common: liver and lung (8.3%), liver and other organs (4.2%), lung and other organs (8.3%), and multi-visceral involvement (4.2%). Oncological redo surgery was performed in 30.6% (15/49).

The survival outcomes described in eight papers [19,27,29,32,35,36,38,45] are summarized in Table 2. The five-year overall survival (OS) ranged from 0 to 51.9%, with an MS of 16–34 months and a DFS of 9–13 months.

Individual survival outcomes could be extracted in 126 patients [16,17,19,20,21,22,23,24,26,27,29,30,32,33,34,35,36,37,40,41,44]. A Kaplan–Meier analysis showed a median OS of 34 months (confidence interval—CI-: 30–40) (Figure 2), and the 1-year, 3-year, and 5-year OS was 71.4%, 19.8%, and 7.1%, respectively (Table 2). The median DFS of individual survival outcomes in 102 patients was 24 months (18–32) (Figure 3). Survival analysis was also performed comparing patients with and without histological invasion of the IVC (*n* = 71), free or affected margins (*n* = 81), and the presence/absence of relapse (local and/or distant) (*n* = 102). Statistical differences were found when comparing the OS of patients with or without relapse (*p* = 0.016) (Figure 4) and the DFS between patients with and without histological invasion of the IVC (*p* = 0.001). However, the OS in patients with or without histological invasion of the IVC (Figure 5 and Figure 6) or with or without affected margins (Figure 7) did not present statistically significant differences.

## 4. Discussion

Liver tumours involving the IVC generally have poor oncological outcomes. The fundamental principle of oncological surgery is the en bloc resection of the tumour and the affected organs. In many of these tumours, low survival rates due to recurrence and limited organ availability have made liver transplantation not a viable option in advanced hepatic tumours with vascular invasion. 

Since the first combined liver and IVC resection described by Starzl in 1980 [46], an increasing number of cases have been published, but a solid review of the literature has not yet been carried out. Most of the studies published so far are case reports or small case series [19,20,24,27,28,29,30,33,34,35,38,39,40,41,43]. The majority are studies that include IVC resections for different types of tumours and analyse the results jointly, meaning that it is impossible to extract reliable results in the subgroup of patients with hepatic metastases of colorectal cancer. 

This review is based on original data and includes the largest number of patients with combined liver and IVC resection for CRLM described to date, thus providing clear evidence of the value of this surgical technique. Considering the complexity of this technique compared with a standard hepatic resection, the postoperative mortality rate of 4.6% can be considered acceptable. Long-term survival is the most important parameter to assess oncological outcomes for malignant tumours. In the case of IVC resection, the 5-year OS rate ranged from 0% [45] to 51.9% [27], rising to a value of 7.1% in this review, the largest as far as we know. Since there is no other effective alternative therapy for these patients, this percentage is also acceptable. The 5-year OS of palliative chemotherapy for CRLM is 2.2% [47]. The survival is comparable with the rates previously described for these tumours [48,49].

The histological study of the resected specimens in the different published studies showed that 40% of hepatic tumours with radiological-suspected IVC involvement had only fibrous adhesion without true tumour infiltration. Distinguishing between inflammatory adhesions and true vascular invasion both pre-operatively and intra-operatively is often impossible. Malde et al. [35] suggested that surgeons should try to peel the tumour from the IVC since, in most cases, this can be achieved. However, there is a high risk of IVC rupture and massive bleeding, as well as tumour spread, although it is true that on many occasions, it is impossible to dissect the tumour from the IVC without producing a defect in the caval wall [26]. Furthermore, the data show that the microscopic invasion of major vessels is comparable to those without it, indicating that invasion of the major vessel is not indicative of aggressive tumour biology [50]. Our results are consistent with these findings, even though DFS is poorer in the presence of IVC histological infiltration. Furthermore, other histological features, such as the state of margins, are not associated with poor OS.

Vascular control was adapted in all the cases during surgery with the objective of minimizing intra-operative bleeding, shortening the ischaemia time, maintaining haemodynamic stability, and improving the tolerance to ischaemia–reperfusion injury of the remnant liver [26,45].

The type of IVC reconstruction depends on the degree of involvement. Usually, if a < 2 cm wall is excised, a direct suture of the IVC is performed. When the vascular involvement is greater than 2 cm, an autologous or synthetic patch is usually used to avoid stenosis. Finally, when the infiltration is at least half the circumference, IVC replacement is required [35].

An autologous vein is the preferred material for IVC replacement to avoid thrombosis and postoperative infection, but it may not be technically feasible, especially if a relatively long segment of IVC needs to be replaced. Therefore, in many cases, reconstructions with prostheses are performed. Infection is an important concern, although no cases have been published in the literature review carried out. In patients in which tumour contact with the IVC is low, it can be controlled by placing tangential vascular clamps to preserve the blood flow.

In those patients in whom tumour involvement requires the resection of a significant segment of the IVC, THVE, which includes continuous clamping of the hepatic hilum and suprahepatic and infrahepatic IVC, with or without a venovenous bypass, is the standard procedure. If the IVC involvement is below the hepatic veins and there is sufficient space to clamp above the tumour, IVC occlusion could be performed below the suprahepatic veins after the hepatic section to minimize the ischaemia time [39]. If there is an involvement of the IVC at the level of the suprahepatic veins, ex situ hypothermic liver resection would be indicated. However, this technique has a high postoperative mortality of up to 28% [51]. An alternative to ex situ resection is the ante situm technique [52]. In this case, after dividing the liver together with the IVC from the retroperitoneum, the liver can be mobilized outside the abdomen. Subsequently, it is perfused with a cold solution through the portal vein, and the parenchyma is resected. Once the resection is completed, reconstruction of the IVC and venous flow is performed. The main advantage of the ante situm procedure over the ex situ procedure is the preservation of the hepatic pedicle, which can significantly reduce postoperative morbidity and mortality [51]. Usually, all these procedures are performed in highly selected patients and in high-volume centres.

This study has several limitations. First, the studies included in this review were retrospective, with an inherent risk of bias. The possible differences in several variables, including the institutions’ and surgeons’ experience, study populations, peri-operative management, surgical procedures, IVC resections, and reconstructions, may have influenced the postoperative and long-term clinical outcomes. Second, the favourable survival rate may be due in part to the fact that patients were highly selected. Third, as the studies included ranged in time more than 30 years, it must be kept in mind that the results may be influenced by the period of time in which the surgery was performed. Fourthly, the survival of patients with CRLM has increased in recent decades, not only because of technically more aggressive surgeries, but also because of the development of new lines of chemotherapy with better disease control. In fact, in most studies, the percentage of patients receiving adjuvant chemotherapy is above 50%, and in some cases, even reaches 80% [53].

## 5. Conclusions

Despite the difficulty of IVC resection, the numerous technical advances recorded in recent years have meant that this procedure is now possible in patients with CRLM involving the IVC. The studies published so far are retrospective, and many of them include resection of the IVC for tumours other than CRLM. In the absence of prospective randomized studies, which are difficult to perform, IVC resection seems to be safe and feasible in experienced centres.

## Figures and Tables

**Figure 1 cancers-15-02965-f001:**
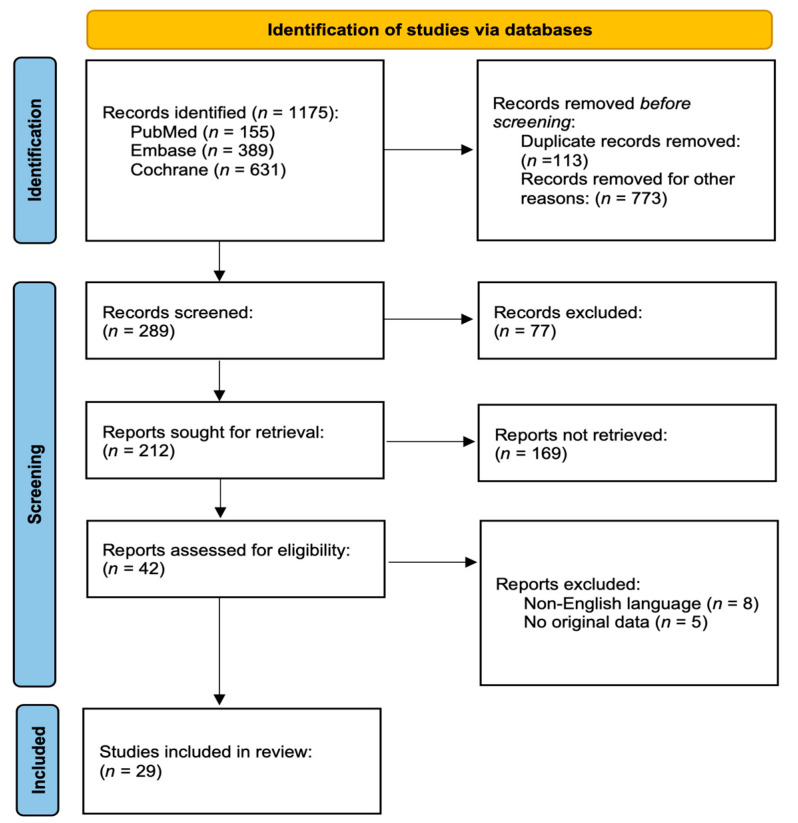
PRISMA 2020 flow diagram.

**Figure 2 cancers-15-02965-f002:**
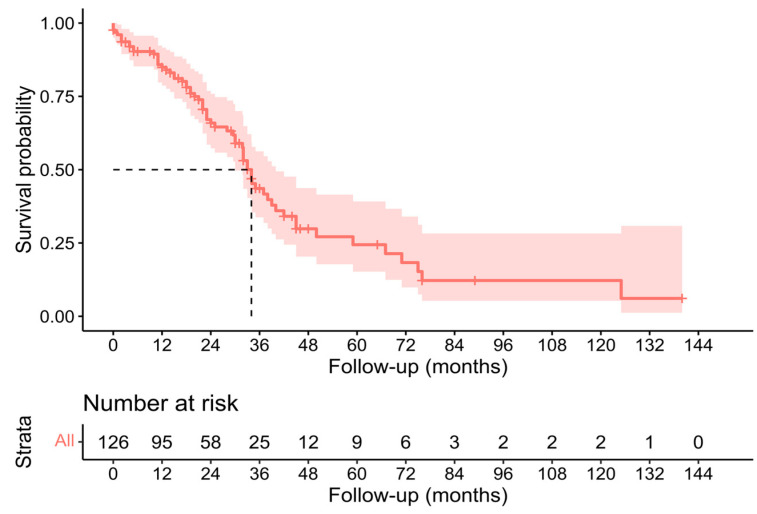
Kaplan–Meier analysis of individual patients’ overall survival outcomes [16,17,19,20,21,22,23,24,26,27,29,30,32,33,34,35,36,37,40,41,44].

**Figure 3 cancers-15-02965-f003:**
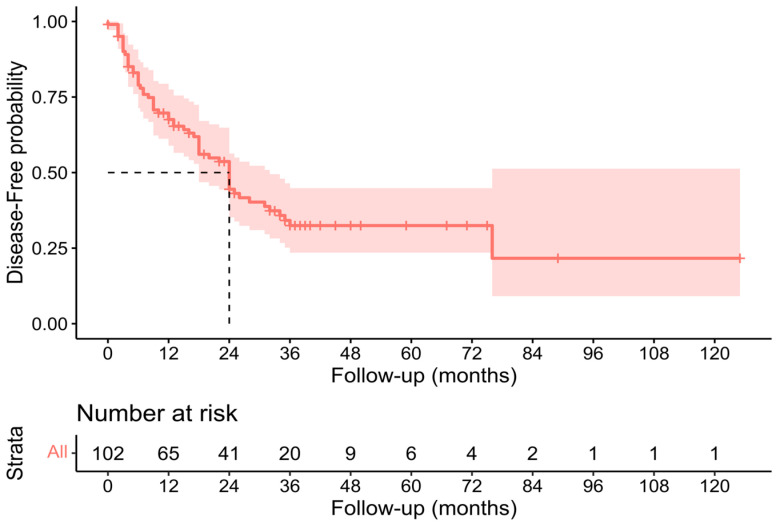
Kaplan–Meier analysis of individual patients’ disease-free survival outcomes [16,17,19,20,21,22,23,24,26,27,29,30,32,33,34,35,36,37,40,41,44].

**Figure 4 cancers-15-02965-f004:**
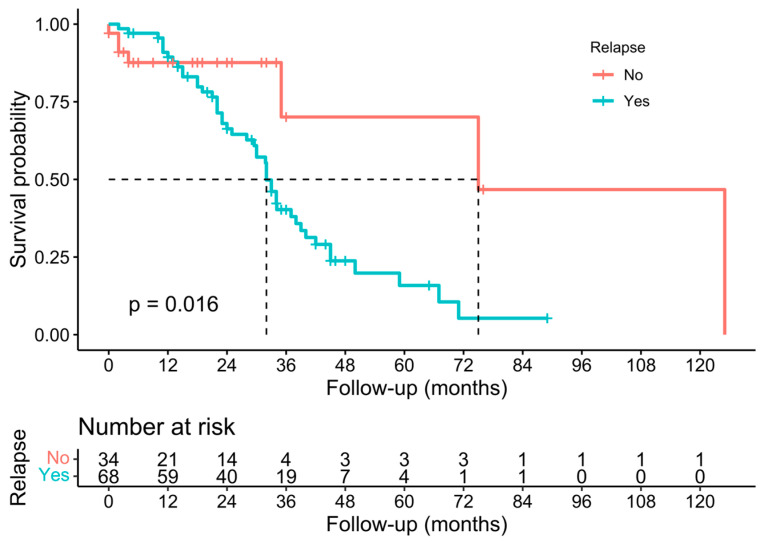
Kaplan–Meier analysis of individual patients’ overall survival outcomes in the presence/absence of relapse [16,17,19,20,21,22,23,24,26,27,29,30,32,33,34,35,36,37,40,41,44].

**Figure 5 cancers-15-02965-f005:**
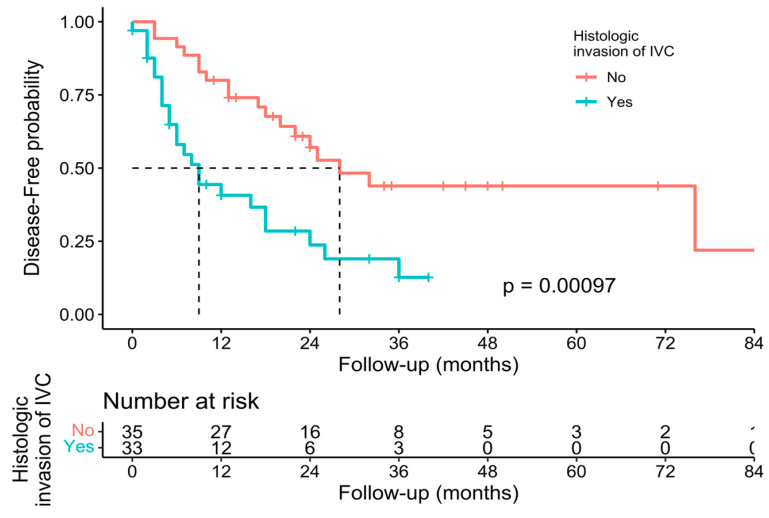
Kaplan–Meier analysis of individual patients’ disease-free survival outcomes comparing presence or absence of histological invasion of the IVC [16,17,19,20,21,22,23,24,26,27,29,30,32,33,34,35,36,37,40,41,44].

**Figure 6 cancers-15-02965-f006:**
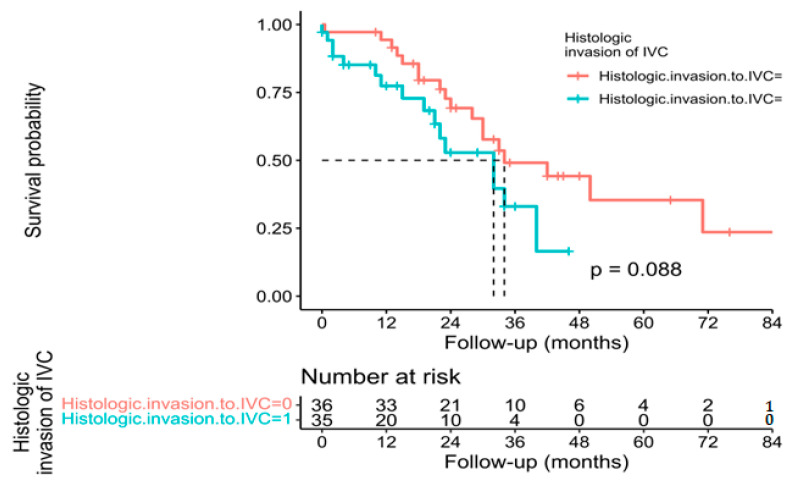
Kaplan–Meier analysis of individual patients’ overall survival outcomes comparing presence or absence of histological invasion of the IVC [16,17,19,20,21,22,23,24,26,27,29,30,32,33,34,35,36,37,40,41,44].

**Figure 7 cancers-15-02965-f007:**
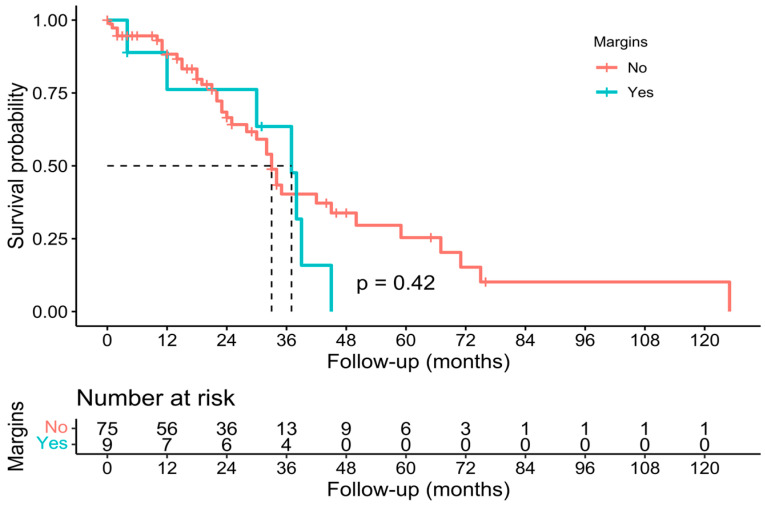
Kaplan–Meier analysis of individual patients’ overall survival outcomes comparing the state of histological margins [16,17,19,20,21,22,23,24,26,27,29,30,32,33,34,35,36,37,40,41,44].

**Table 1 cancers-15-02965-t001:** Articles included in the systematic review.

Year	Author	Country	N° Pat.	Mean Age	M/F	MNM	Type of Hepatectomy	Vascular Control	IVC Repair
1992	Gardner [16]	USA	1	77	0/1	1	1 S1 resection	1 THVE	1 PC
1994	Del Campo [17]	Canada	1	64	1/0	4	1 RH	1 IVC clamp	1 PBP
1997	Kubota [18]	Japan	1	59	0/1	ND	1 RH	1 LC	1 Aut. patch
1999	Miyazaki [19]	Japan	14	ND	ND	ND	ND	ND	ND
2000	Tono [20]	Japan	3	62.3	2/1	1.3	2 extended LHs + CL 1 S1 resection	3 LC	3 PCs
2000	Lodge [21]	UK	8	62	6/2	1.6	4 ex vivo hepatectomies 3 RHs 1 extended RH + CL	2 Portosystemic venovenous bypasses 2 Portosystemic venovenous bypasses + HVE 2 Systemic venovenous bypasses + HVE 1 LC 1 IVC clamp	3 Dacron patches 1 Dacron tube 2 PTFE tube grafts 1 PC 1 Aut. graft
2002	Togo [22]	Japan	1	53	0/1	5	1 LH + CL	IVC clamp	1 Aut. patch
2003	Arii [23]	Japan	2	57.5	1/1	1	1 RT + CL 1 RH	1 Active ventriculovenous shunt (biopump) 1 Passive ventriculovenous shunt	2 PTFE grafts
2004	Aoki [24]	Japan	3	60.6	1/2	3	2 extended RHs 1 wedge resection	3 LC	1 Aut. graft 2 PCs
2004	Hemming [25,26]	USA	6	ND	ND	2.75	3 RTs 1 LT. 1 Ex vivo hepatectomy 1 RH + CL	ND	4 PTFE tube grafts 1 PTFE patch 1 PC
2005	Nardo [27]	Italy	11	55.3	8/3	2.2	4 extended LHs 4 extended RHs 1 RH 1 LH 1 segmentectomy	6 THVEs 1 IVC clamp 2 IHPCs 2 IHPCs/LC	7 PCs 3 PTFE grafts 1 PBP
2005	Yamamoto [28]	Japan	4	61.2	3/1	NA	2 S1 resections 2 RHs + CL	3 LC 1 IVC clamp	3 PCs 1 Aut. graft
2006	Johnson [29]	Canada	11	60.5	4/7	1.1	3 RTs 3 RHs 2 LHs 1 RH + COR 1 RT + COR 1 LT+ COR	10 THVEs 1 LC	8 PTFE patches 2 PCs 1 PTFE tube graft
2007	Delis [30]	USA	6	46.2	3/3	ND	4 RHs 2 LTs	3 THVEs 3 PHVEs	6 PTFE tube grafts
2007	Varma [31]	France	1	66	1/0	ND	ND	1 IVC clamp	1 PC
2008	Hashimoto [32]	Japan	15	62	9/6		2 extended RHs 1 right lateral sectionectomy 1 extended LH 9 wedge resections 1 S1 resection 1 segmentectomy	14 LC 1 IVC clamp	12 PCs 2 Aut. grafts 1 PTFE patch
2009	Kim [33]	Korea	1	56	1/0	1	1 segmentectomy + CL	1 THVE	1 Aut. graft
2009	Tanaka [34]	Japan	20	63.4	10/10	3.7	ND	17 LC 3 THVEs	17 PCs 2 PTFE grafts 1 Aut. patch
2011	Malde [35]	UK	21	57.9	11/10	ND	5 RHs 4 RTs 4 RTs + CL 3 LHs 1 LH + COR 1 LT + CL 1 S1 + COR 1 RH + CL 1 segmentectomy	13 THVEs 4 THVEs, IHP, ex vivo 1 THVE, IHP, ante situm	14 PCs 3 Dacron grafts 2 PTFE grafts 2 PBP
2011	Nuzzo [36]	Italy, Spain	11	55.3	6/5	ND	3 RHs 6 extended RHs 2 extended LHs	ND	7 PCs 4 PTFE grafts
2013	Polistina [37]	Italy	2	55	1/1	2.5	1 extended RH 1 extended LH	2 THVEs	2 Aut. patches
2013	Ie [38]	Japan	9	55	4/5	ND	6 RHs 2 segmentectomies 1 wedge resection	9 LC	9 PCs
2013	Hemming [39]	USA	13	ND	9/4	ND	5 RTs 5 RHs 2 LTs 1 LH	ND	9 PTFE grafts 2 ND patches 2 PCs
2015	Guerrini [40]	Italy	1	67	1/0	1	1 segmentectomy	1 THVE	1 PTFE graft
2015	Marangoni [41]	UK	1	52	0/1	1	1 segmentectomy + CL	1 THVE	1 PTFE patch
2015	Nomi [42]	France	1	58	0/1	ND	1 RH + CL	ND	1 PC
2016	Ko [43]	Japan	6	ND	ND	6.3	2 extended RHs 1 segmentectomy 2 RHs 1 wedge resection	6 THVEs	ND
2016	Tardu [44]	Turkey	1	66	0/1	5	1 RH + COR	1 THVE	1 Aut. graft
2021	Vladov [45]	Bulgaria	13	58.8	5/8	ND	7 RHs 2 RH + CL 1 LH + wedge resection 3 segmentectomies	ND	11 PCs 1 PTFE graft 1 Dacron patch

ND: not determined. RH: right hepatectomy. LH: left hepatectomy. RT: right trisegmentectomy. LT: left trisectionectomy. COR: combined organ resection. CL: caudate lobe. THVE: total hepatic vascular exclusion. PHVE: partial hepatic vascular exclusion. LC: lateral clamping. PC: primary closure. Aut.: autologous. PBP: patch of bovine pericardium.

**Table 2 cancers-15-02965-t002:** Summary of survival outcomes.

Year	Author	*n*	1 Year-OS	3 Year-OS	5 Year-OS	MS (Months)	Median DFS (Months)
1999	Miyazaki [19]	14	64%	33%	22%		
2005	Nardo [27]	11	81.8%	-	51.9%		
2006	Johnson [29]	11				34	9
2008	Hashimoto [32]	15	52.9%	47.3%	-		
2011	Malde [35]	21	75.9%		19.6%	28	-
2011	Nuzzo [36]	11				16	
2013	Ie [38]	9	100%	60%	40%		13 (3–114)
2021	Vladov [45]	13	46%	23%	0%		
2023	Present study	126	71.4%	19.8%	7.1%	34 (30–40)	24 (18–32)

OS: overall survival; MS: median survival; DFS: disease-free survival.

## Data Availability

Reported results can be fount at https://www.researchregistry.com/browse-the-registry#registryofsystematicreviewsmeta-analyses/registryofsystematicreviewsmeta-analysesdetails/6435435f3a902d00281da626/ (accessed on 18 May 2023).

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
