# Peer review of "Surgery of Colorectal Liver Metastases Involving the Inferior Vena Cava: A Systematic Review"

_cancers, 2023, doi:10.3390/cancers15112965_

Round 1

Reviewer 1 Report

Major points;

1) Table 1; All clinical information is lacking in Miyazaki et al.'s report (19) and Azoulay et al.'s report (39). Why were these reports included in the present analysis? These reports account for 40 cases, 18.5% of the 216 cases in this paper, and 31.7% of the 126 cases for which survival information is clear (Table 2).  So, the impact is likely to be significant.

2) Are cases with extrahepatic metastasis at the time of surgery excluded? It greatly affects survival.

3) Table 2; The present study is a summary that includes the 9 studies listed above it, so it should not be listed in the same style in one table as well.

Author Response

Thank you very much for the opportunity to make changes to the manuscript. We appreciate the time and effort you and each of the reviewers have dedicated in providing insightful feedback on ways to strengthen our paper. We have revised the manuscript in accordance with the reviewer’s suggestions. Our changes have been highlighted in the revised manuscript.

Our point-by-point responses to the reviewer’s comments and the revisions made are summarized below in the text. The reviewers’ comments helped us considerably in improving our manuscript.

In the same way, we have adapted the text according to the publication standards of Cancers.

Reviewer 1:

  • Table 1: All clinical information is lacking in Miyazaki et al.'s report (19) and Azoulay et al.'s report (39). Why were these reports included in the present analysis? These reports account for 40 cases, 18.5% of the 216 cases in this paper, and 31.7% of the 126 cases for which survival information is clear (Table 2). So, the impact is likely to be significant.

Azoulay et al.’s report has not been taken into account, following the instructions of Revisor 1.  Miyazaki et al.’s [19] report describe clinical information of all the patients undergoing a combined liver and IVC resection for all kinds of cancer. Specific information of colorectal liver metastases could not be obtained. However, specific survival outcomes of these groups are described, obtaining a valuable data that is just reflected in another seven studies [27,29,32,35,36,38,45]. That is why this paper is included [19]. The data resulting from the exclusion of the Azoulay’s study have been modified, highlighted in yellow throughout the text.

  • Are cases with extrahepatic metastasis at the time of surgery excluded? It greatly affects survival.

No extrahepatic metastasis at the time of surgery are clearly specified in any of the patients.

  • Table 2: The present study is a summary that includes the 9 studies listed above it, so it should not be listed in the same style in one table as well.

The present study’s survival outcomes analysis include Nardo et al. [27], Johnson et al. [29], Hashimoto et al. [32], Malde et al. [35] and Nuzzo et al. [36] reports, previously reflected in Table 2. In these papers, individual survival outcomes were described, allowing a global survival analysis. These summarized results are much more consistent than separately reported. And, in our opinion, it is much easier to compare the information obtained when searched in the table.

Reviewer 2 Report

Title: Surgery of colorectal liver metastases involving the inferior vena cava: a systematic review

This paper describes patients who underwent surgery for CRLM involving IVC as well as evaluation of the safety and clinical and oncological long-term outcomes.

The manuscript cites many references and is well written. However, there are some questions and the author is requested to add the descriptions according to comments as below.

1. Patients

What is the detail about morbidity and mortality outcomes ?

The authors should describe about grade 3a in four and grade 5 complication?

2. Overall mortality

Overall mortality was 48.2% (66/137), extremely high. The authors should describe about

the causes of mortality.

Author Response

Thank you very much for the opportunity to make changes to the manuscript. We appreciate the time and effort you and each of the reviewers have dedicated in providing insightful feedback on ways to strengthen our paper. We have revised the manuscript in accordance with the reviewer’s suggestions. Our changes have been highlighted in the revised manuscript.

Our point-by-point responses to the reviewer’s comments and the revisions made are summarized below in the text. The reviewers’ comments helped us considerably in improving our manuscript.

In the same way, we have adapted the text according to the publication standards of Cancers.

Reviewer 2:

  • Patients What is the detail about morbidity and mortality outcomes? The authors should describe about grade 3a in four and grade 5 complication?

The following paragraph has been added: “Grade 3a complications included: three cases of pleural effusion that required surgical drainage [21], and one case that required continuous veno-venous hemofiltration and endovascular treatment because of a renal artery stenosis [37]. Grade 5 complication (in-hospital mortality) was specified in seven patients [21,27,35,36,39]. The causes were liver failure in three cases [27,35,36], multiorgan failure in another two cases [21,35], hemopneumothorax, renal and respiratory failure in one case [21] and pulmonary embolism in the remaining [39]”. Page 7, Morbidity and mortalitity outcomes.

  • Overall mortality: Overall mortality was 48.2% (66/137), extremely high. The authors should describe about the causes of mortality.

The following sentence has been added: “Overall mortality rate (including in-hospital mortality) raises 48.2%. A total of 46 patients (69.7%) died of disease progression. Two patients died with evidence of disease [36,37]. No information could be obtained in the remaining eleven patients”. Page 7, Morbidity and mortalitity outcomes.

Round 2

Reviewer 1 Report

Questions are well answered.

Reviewer 2 Report

Title: Surgery of colorectal liver metastases involving the inferior vena cava: a systematic review

This paper describes patients who underwent surgery for CRLM involving IVC as well as evaluation of the safety and clinical and oncological long-term outcomes.

The manuscript cites many references and is well written. Author`s responses to the reviewer's comments were thorough and well-written. I believe this manuscript is acceptable.